# Deciphering the interactome of Ataxin-2 and TDP-43 in iPSC-derived neurons for potential ALS targets

Yuan Tian[1]*, Nicolette Heinsinger[1], Yinghui Hu[1], U-Ming Lim[2], Yi Wang[1], Aaron Zefrin Fernandis[2], Sophie Parmentier-Batteur[1], Becky Klein[1], Jason M. Uslaner[1], Sean M. Smith[1]

1 Neuroscience, Merck Research Laboratories, Merck & Co., Inc., Rahway, New Jersey, United States of America, 2 Quantatitive Biosciences, Merck Sharp & Dohme, Singapore, Singapore

* yuan.tian1@merck.com

**Data Availability Statement:** All relevant data are within the manuscript and its Supporting Information files.

## Abstract

Ataxin-2 is a protein containing a polyQ extension and intermediate length of polyQ extensions increases the risk of Amyotrophic Lateral Sclerosis (ALS). Down-regulation of Ataxin-2 has been shown to mitigate TDP-43 proteinopathy in ALS models. To identify alternative therapeutic targets that can mitigate TDP-43 toxicity, we examined the interaction between Ataxin-2 and TDP-43. Co-immunoprecipitation demonstrated that Ataxin-2 and TDP-43 interact, that their interaction is mediated through the RNA recognition motif (RRM) of TDP-43, and knocking down Ataxin-2 or mutating the RRM domains rescued TDP-43 toxicity in an iPSC-derived neuronal model with TDP-43 overexpression. To decipher the Ataxin-2 and TDP-43 interactome, we used co-immunoprecipitation followed by mass spectrometry to identify proteins that interacted with Ataxin-2 and TDP-43 under conditions of endogenous or overexpressed TDP-43 in iPSC-derived neurons. Multiple interactome proteins were differentially regulated by TDP-43 overexpression and toxicity, including those involved in RNA regulation, cell survival, cytoskeleton reorganization, protein modification, and diseases. Interestingly, the RNA-binding protein (RBP), TAF15 which has been implicated in ALS was identified as a strong binder of Ataxin-2 in the condition of TDP-43 overexpression. Together, this study provides a comprehensive annotation of the Ataxin-2 and TDP-43 interactome and identifies potential therapeutic pathways and targets that could be modulated to alleviate Ataxin-2 and TDP-43 interaction-induced toxicity in ALS.

## Introduction

TAR DNA-binding protein of 43 kDa (TDP-43) plays a critical role in RNA-related metabolism [1]. In Amyotrophic Lateral Sclerosis (ALS) and Frontal Temporal Lobe Dementia (FTLD) patients, inclusion bodies are often observed in affected regions in the central nervous system (CNS), including the hippocampus, neocortex, and spinal cord, with deposition of TDP-43 aggregates [2]. It was reported that TDP-43 toxicity included both loss of physiological

**Funding:** The author(s) received no specific funding for this work.

**Competing interests:** All authors were employees of Merck & Co., Inc., Rahway, NJ, USA at the time of this study. Employment does not alter authors' adherence to the journal's policies on conflicts of interest or sharing data and materials.

functions and gain of toxic functions. Ataxin-2 is a polyglutamine (polyQ) protein in which long (>34) polyQ expansions cause spinocerebellar ataxia 2 (SCA2) [3–5] and intermediate-length (27–33) repeats are a risk factor for ALS [6, 7].

Ataxin-2 encoded by the gene *ATXN2*, has emerged as a therapeutic target because it is a potent genetic modifier of TDP-43 toxicity and aggregation [6, 8]. Increased levels of Ataxin-2 exacerbate TDP-43-induced toxicity, while decreased Ataxin-2 levels mitigate the toxicity [9–11]. It was suggested that Ataxin-2 modifies TDP-43 toxicity through their interaction at stress granules where many RNA binding proteins genetically implicated in ALS risk localize including TDP-43, Ataxin-2, FUS, TIA1 and TAF15 [12]. Further implicating Ataxin-2 in ALS, antisense oligonucleotides (ASOs) or RNAi targeting ATXN-2 *mRNA* provide marked protection against motor deficits in mouse models of TDP-43 proteinopathy [9, 11] and SCA2 [13]. These results have motivated recent administration of ASOs targeting *ATXN2* to human patients with ALS in a phase 1 clinical trial (ClinicalTrials.gov: NCT04494256). Despite their enormous potential, questions remain regarding the safety or dosage limitations of using ASOs, and little is known about how Ataxin-2 is normally regulated for TDP-43 toxicity and what other co-factors are involved in TDP-43 pathology.

We have previously reported an iPSC-derived neuronal ALS model by overexpressing TDP-43 via Adeno-associated virus (AAV) infection. This neuronal model manifests the disease phenotypes of neuronal loss, TDP-43 phosphorylation and mis-localization [14]. Using this model, we examined the interaction of Ataxin-2 with TDP-43 by co-immunoprecipitation and showed their interaction is mediated through the RRM domains of TDP-43. We further demonstrated Ataxin-2 and its interaction with TDP-43 was involved in TDP-43 toxicity and stress granule formation in the iPSC-derived neuronal ALS model. Using co-immunoprecipitation followed by mass-spectrometry, we profiled the Ataxin-2 and TDP-43 interactome under the conditions of endogenous or overexpressed TDP-43 in iPSC-derived neurons. Many proteins were differentially regulated by TDP-43 overexpression and toxicity, including those involved in RNA regulation, cell survival, cytoskeleton reorganization, protein modification, and diseases. Interestingly, the RNA-binding protein (RBP), TAF15 which has been implicated in ALS [15], was identified as a strong binder of Ataxin-2 in the condition of TDP-43 overexpression. In summary, this study sheds light on the interaction between Ataxin-2 and TDP-43 and its involvement in TDP-43 toxicity. Our comprehensive annotation of Ataxin-2 and TDP-43 interactome revealed promising therapeutic targets that could be modulated to ameliorate TDP-43 toxicity for the treatment of ALS.

## Material and methods

### Animals and tissue collection

The Tar6/Tar6 animal line was generated and described by Wils H. et al. [16]. A mouse colony was maintained at Taconic Biosciences (Rensselaer, NY). Animals were acclimated to housing for a week or longer prior to any studies. Access to food and water were provided *ad libitum* and the animals were kept on a normal 12 hour:12 hour light:dark cycle. All animal studies were performed under the approval of the Institutional Animal Care and Use Committee of Merck & Co., Inc., Kenilworth, NJ, USA. For brain tissue collection, animals were euthanized by carbon dioxide and transcardially perfused with heparinized phosphorylated buffered saline (PBS; 3U/mL).

### Mouse brain tissue lysates prep

Brain cortex samples for immunoprecipitation were collected from WT or Tar6/6 mice. Samples were homogenized on TissueLyser II (Qiagen, 85300) in IP lysis buffer (50 mM Hepes,

150 mM NaCl, 1 mM EDTA, 1 mM EGTA, 10% glycerol, 1% Triton X-100, 25 mM NaF, 10 uM ZnCl2; pH 7.5 plus 1x Halt Protease and Phosphatase Inhibitor Cocktail (ThermoFisher, 78447)) with sample:buffer ratio of 1:5. The samples were then centrifuged for 15 min at 16,000×g, + 4˚C. Supernatants were collected and stored at − 80˚C pending analysis.

## Immunoprecipitation

Dynabeads Protein A Immunoprecipitation Kit (ThermoFisher, 10006D) was used for immunoprecipitation analysis. 5ug antibody was linked to 1.5mg Dynabeads overnight at 4˚C according to manufacture protocol. Antibodies used were Anti-TDP-43 (Proteintech, 10782-2-AP), Anti-Flag (Sigma, F1804), Normal Rabbit IgG (Sigma, 12–370), Anti-Ataxin-2 (Proteintech, 21776-1-AP). 450ul cell lysates or brain lysates were added to each antibody-conjugated beads, incubated with rotation at 4˚C overnight. Dynabeads-Ab-Ag complex was then washed three times and eluted in 30ul Elution Buffer with 2 minutes incubation at room temperature. 10ul 4x LDS sample buffer (Invitrogen, NP0007) was added to each sample for Western blot analysis.

## Western blot and quantification

Samples in 1x LDS sample buffer (Invitrogen, NP0007) were loaded on NuPAGE 4–12% Bis-Tris gel (Invitrogen, WG1403BOX) and run at 200V for 40 minutes in 1x MES running buffer (Invitrogen, NP0002). The gel was transferred on Trans-Blot Turbo Transfer system (BioRad) to 0.2μm PVDF membrane (BioRad, 1704157). The membrane was blocked with Intercept (TBS) Blocking Buffer (Li-cor Biosciences, 927–60001) for 1 hour and probed with primary antibodies in antibody dilution buffer (0.1% Tween-20 in TBS blocking buffer) overnight at 4˚C. Primary antibodies used were: Anti-TDP-43 (Proteintech, 10782-2-AP), Anti-Flag (Sigma, F1804), Anti-Ataxin-2 (Proteintech, 21776-1-AP). The membrane was washed three times in TBST buffer and incubated for 1 hour with 1:10,000 dilution of IRDye secondary antibodies (Li-cor Biosciences, 926–68072, 926–32213) in antibody dilution buffer. The membrane was washed three times in TBST, visualized using the Odyssey imaging system (Li-cor Biosciences). Image Studio software (Li-cor Biosciences) was used to quantify western blot imaging results. Intensity of bands was normalized to the beta-actin, and then a one way ANOVA was performed using GraphPad Prism.

## HEK297 transfection and proximity AlphaLISA

HEK293FT cells (Thermo Fisher #R700-07) were co-transfected with plasmid DNA pcDNA3.1 constructs of TDP-43-flag (or TDP-43mutRRM-flag) together with *ATXN2* (or polyQ *ATXN2* variants), using Lipofectamine 2000 Reagent (Invitrogen #11668–027) according to manufacture protocol. 72hr post transfection, cells were lysed with AlphaLISA lysis buffer (PerkinElmer #AL003). AlphaLISA proximity assay was then used to measure the interaction between Ataxin-2 (recognized by biotinylated Ataxin-2 antibody/streptavidin donor beads) and the flag-tagged TDP-43 (recognized by flag-acceptor beads). 10ul cell lysates were incubated at room temperature for 1 hour with 10ul 20ug/ml anti-FLAG Acceptor beads (PerkinElmer, #AL112C) plus 10ul biotinylated Ataxin-2 Rabbit anti-Human Polyclonal Antibody (LSBio, #LS-C501676-100). 10ul 40ug/ml Streptavidin Donor beads (PerkinElmer, #6760002) were then added to the mixture. After 1 hour room temperature incubation, AlphaLISA signal was read on Envision plate reader (PerkinElmer).

## TaqMan qPCR

After treatment in 96-well plates, cells were washed once with PBS, followed by the addition of 30ul/well lysis buffer (ABI, 4448536). Reverse transcription was carried out with High-capacity cDNA RT kit (ABI, 4368813) with 1/10 volume of cell lysates following the manufacturer's instructions. 3ul of resulting cDNA was subjected to TaqMan quantitative PCR using TaqMan Fast Advanced Mix (ABI, 4444558) in 10ul total volume. TaqMan Gene Expression Assays used in the TaqMan qPCR were Human *ATXN2* (ABI, Hs00268077_m1) Human TAF15 (Hs00896645_m1) and internal control human Ppib (ABI, Hs00168719_m1). Real-time PCR reactions were performed on the ViiA 7 Real-time PCR system (ABI) using the following thermocycling conditions: 2 min at 50˚C, 20 sec at 95˚C, followed by 40 cycles of 1 sec at 95˚C and 20 sec at 60˚C. Relative gene expression levels were calculated using the ΔΔCt method.

## iPSC derived GABA neuron culture, AAV and lentivirus (LV) infection, cell lysate prep, and immunocytochemistry (ICC)

Human iPSC derived GABANeurons (Cellular Dynamics International, Cat. No. NRC-100-010-001) were cultured in iCell neuron maintenance media (Cat No. NRM-100-121-001) with iCell neuron supplement media (Cat No. NRM-100-031-001) according to manufacturer's instructions. In brief, the cells were differentiated into human IPSC-derived cortical neurons from the fibroblasts of a young, healthy Caucasian female by CDi. Cell vials were thawed in a 37 degree C water bath for 2–3 mins, and resuspended slowly in 10mL media. Cells were centrifuged at 300g for 5 mins, and resuspended in fresh media. Cells were plated at 40k/well density in 200ul media on 96-well black poly-D lysine pre-coated plates (Greiner, Cat. No. 655946), with 1:300 dilution of laminin (1mg/ml, Sigma, Cat. No. L2020-1MG) added to the media. Wells were given a half media change 2x per week. On DIV4, cells were transduced with Lentivirus shRNA (ATXN-2 clone ID TRCN291127, Sigma-Aldrich; TAF15 clone ID TRCN0000285486, Sigma) at 3 million GC/cell. On DIV5, cells were transduced with AAV (AAV8-syn-Ctrl, AAV8-syn-TDP-43, or AAV8-syn-TDP-43-mutRRM) at one million GC/cell. On DIV13, cells were lysed in IP lysis buffer (50 mM Hepes, 150 mM NaCl, 1 mM EDTA, 1 mM EGTA, 10% glycerol, 1% Triton X-100, 25 mM NaF, 10 uM ZnCl2; pH 7.5 plus 1x Halt Protease and Phosphatase Inhibitor Cocktail (ThermoFisher, 78447)) for subsequent immunoprecipitation assay. On DIV 17 or day 12 post-AAV, cells were fixed with 4% PFA for ICC staining. To induce stress granule formation, cells were treated with 0.25mM sodium arsenite for 1hr, then fixed with 4% PFA for ICC. For ICC, cellular nuclei were stained using 2 drops/mL of NucBlue Live cell stain (Life Technologies Cat. No. R37605) and incubated for 20–25 mins. Media was removed and replaced with 100 ul of new media. Cells were fixed with 4% formaldehyde for about 10 mins at room temperature. Cells were then incubated in blocking buffer (2% FBS, 2% Donkey serum, 0.2% Triton X-100 diluted in DPBS) for >15 mins at room temperature. Primary antibodies were incubated in blocking buffer overnight at 4C. The next day, cells were washed in wash buffer (2% FBS diluted in DPBS) and secondary antibodies were incubated in wash buffer for >2 hours at room temperature. Plates were imaged using ArrayScan (ThermoFisher Scientific) and analyzed using the neuroprofiling algorithm for MAP2 (neuronal survival, morphology analysis) or the compartment analysis for TDP-43 expression and mislocalization. Data were graphed with Prism 7 and analyzed via one way ANOVA.

Primary antibodies used include anti-Map2 (Millipore, Cat No. Mab3418), anti-cleaved caspase-3 (Asp175) (Cell signaling, Cat. No. 9661L), Anti-TDP-43 (Proteintech, Cat. No. 10782-2-AP, rabbit poly Ab), anti-eIF3h(C-5) (Santa Cruz, Cat. No. sc-137214, mouse mAb), Anti-Ataxin-2 (BD Bioscience Cat. No. 611378, mouse mAb). Secondary antibodies used

include Alex Fluor 488 goat anti-mouse (life technologies, Cat. A11029), Alexa Fluor 647 goat anti-rabbit (Life technologies, Cat. A21245).

## High content imaging and analysis

High-content imaging was performed to acquire images using an ArrayScan (ThermoFisher) with 20X and 40X objectives before high-content imaging analysis. To cover a good portion of the surface of the well, ≥9 fields per well were imaged, analyzed, and averaged using 20X-magnification images to increase the number of cells analyzed. A total of 6 wells of a 96-well plate were used per group ($n = 6$), and each experiment was repeated multiple times to ensure reproducibility. Image analyses and calculations were performed using Cellomics software (ThermoFisher). Hoechst staining was used to label cell nuclei. Single-cell identification was performed using the Hoechst nuclei stain. Within the cell mask per each experiment, we quantified Ataxin-2 or TDP-43 signal within the nuclei stain and the cytoplasm, which was defined as a constant pixel distance surrounding the nuclei stain. Nuclei were counted, and 9 fields were averaged across wells to assess cell viability. To determine the changes in protein expression, a compartmental analysis algorithm (Cellmoics) was utilized. In brief, "spots" were identified if they presented a fluorescent intensity higher than a defined threshold. The threshold was established based on the distribution of the fluorescence intensity measured in EV-transfected cells without the expression of designated proteins. Defined thresholds were calibrated, and standardized the analyses based on this background fluorescence. Results were expressed as the total intensity of "spots" per well and then averaged across 6 wells per all groups.

## IP protein extractions and trypsin digestion

For each IP sample, protein content was solubilized from the Dynabeads with 50μL of lysis buffer containing 4% sodium dodecyl sulfate, 50 mM Tris-HCl pH 7.5, 50 μg/mL of DNase (Roche, 10104159001, 50 μg/mL of RNase (Roche, 10109169001) and HALT$^{TM}$ protease inhibitor cocktail (Thermo Scientific, 78430). Detergent removal and protein digestion were performed in centrifugal suspension trap columns (Protifi, C0$_2$-micro) according to manufacturer's instructions. Briefly, each of the IP samples was reduced with 50 mM of dithiothreitol (Sigma, 43815) at 95˚C for 10min and then alkylated with 100 mM of iodoacetamide (Sigma, I2512) in the dark at ambient temperature for 30min. The samples were acidified with 1.2% phosphoric acid and mixed well with wash buffer consisting of 90% methanol and 100 mM ammonium bicarbonate. The samples were transferred to the suspension trap columns and centrifuged at 4000 g for 30 s. The columns were washed 3 times with wash buffer. Proteins trapped in the suspension bed were digested in 50 mM ammonium bicarbonate with trypsin and endoproteinase Lys-C (Promega, V5073) at enzyme to protein ratio 1:25 in a 37˚C waterbath overnight. Peptides were eluted firstly with 50 mM ammonium bicarbonate, then with 0.2% formic acid and lastly with 50% acetonitrile and 0.2% formic acid. The elutes were pooled and vacuum dried completely for mass spectrometry analysis.

## Mass spectrometry analysis

LC-MS analysis was conducted using an Sciex 6600+ TripleTOF mass spectrometer (Concord, Ontario, Canada) coupled to an Eksigent$^{TM}$ NanoLC 425 system (Dublin, CA). For IDA spectral library construction, peptides were trapped on a NanoLC pre-column (Chromxp C18-LC-3μm, size 0.35 × 0.5 mm, Eksigent), and separated on an analytical column (C18-CL-120, size 0.075 × 150 mm, Eksigent) using a 120-min gradient from 5 to 35% Buffer B (Buffer A: 2% ACN, 98% H2O, Buffer B: 98% ACN, 2% H2O, 0.1% FA) at a flow rate of 300 nL/min. Full-scan MS was performed in positive ion mode with a nano-ion spray voltage of 2.1 kv from 350

to 1500 (m/z). Up to 50 precursors were selected for MS/MS (m/z 100–1500), based on inclusion criteria of an intensity greater than 150 counts/s, a charge state from + 2 to + 5, a mass tolerance of 50 mDa and dynamic exclusion for 15 s. Ions were fragmented in the collision cell using rolling collision energy. For SWATH™ acquisition, the same chromatographic conditions used in the IDA runs and a set of 55 overlapping windows (400–1250 Da) was constructed with variable collision energies appropriate for a 2+ ion centered in the window with a spread of 15 eV.

## Generating the reference spectral library

For each bio-replicate analysis, two IDA injections were performed to increase protein coverage and the MS files were searched using ProteinPilot software (Version 5.1, Sciex) with the Paragon algorithm. The samples were processed as unlabeled samples using the following parameters: cysteine alkylation with iodoacetamide, trypsin digestion, and no special factors. The ID search was performed using the human UniProt database (May 2019 release).

## SWATH-MS data analysis

For the DIA samples, spectral alignment and targeted data extraction were carried out using the SWATH Processing Micro App in Peakview software (Version 1.2, Sciex). These analyses were performed using the reference spectral library that was previously generated. For each comparison group, four DIA raw files were simultaneously loaded with an extraction window of 20 minutes. The parameters used for data analysis included the selection of 5 peptides, 8 transitions, and a minimum peptide confidence level of 99%. These parameters considered shared peptides and set the XIC (extracted ion chromatogram) width at 50 ppm. Following data processing, the processed mrkvw files that contained protein information obtained from PeakView were imported into MarkerView software (Version 1.2.1, Sciex). The protein intensities (peak areas) for all runs were normalized using the built-in total ion intensity sum plug-in. Prior to subsequent statistical analysis, a Log2 transformation was applied to the data. The adjusted p-values for each protein were calculated using an unpaired t-test, without assuming a consistent standard deviation. To control false discoveries, a two-stage step-up method of Benjamini, Krieger, and Yekutieli with a Q-value of 5% was employed. Differentially expressed proteins are defined by Log2 (Fold change) >1 and -Log10 (adjusted P) >1.301.

## Ingenuity pathway analysis

All TDP-43 interacting proteins of the TDP-43 vs Normal comparison group were separately analyzed in the IPA software version 24.0.1 (QIAGEN Inc.). The gene symbols and corresponding expression fold change values of the identified proteins were inputted into IPA (Ingenuity Pathway Analysis). The Core Analysis function was executed with the following parameters: fold change in expression was selected as the analysis type, networks were generated considering both direct and indirect relationships, and the confidence level was restricted to experimentally observed data for the relationships. The cut-off values applied to all datasets included fold change ≥1.5 for up-regulated and ≤-1.5 for down-regulated proteins. Adjusted p values (Benjamini-Hochberg, FDR) of <0.05 were considered significant. Based on the IPA's analysis, significant canonical pathways, biological functions and diseases, and interaction networks were algorithmically generated.

## Results

### Ataxin-2 interacts with TDP-43 in both iPSC-derived GABA neurons and mouse brain tissue

We first aimed to establish the interaction between Ataxin-2 and TDP-43 in vitro and in vivo. To do this, we overexpressed human TDP-43 in human iPSC-derived GABA neurons (iCell neurons) using scAAV8-Syn-TDP-43, and immunoprecipitated the lysates with IgG or an Ataxin-2 antibody. We then probed the elutes with either Ataxin-2 or TDP-43 via Western blot. We found a robust expression of both Ataxin-2 and TDP-43 in the Ataxin-2 antibody immunoprecipitants but not in the IgG control, indicating an interaction between Ataxin-2 and TDP-43 protein in iPSC neurons (Fig 1A).

We next asked if this interaction between Ataxin-2 and TDP-43 occurs *in vivo*. Wild type (WT) and Tar6/Tar6 mouse brain lysates from 6 month old animals were also immunoprecipitated with either IgG control or Ataxin-2 antibody and probed for Ataxin-2 and TDP-43 via Western blot. Tar6/Tar6 is a transgenic mouse line that overexpresses human wild-type TARDBP (encodes for TDP43) at about 1.2-fold over endogenous protein level [16]. In both WT and Tar6/Tar6 mice, we observed co-immunoprecipitation of Ataxin-2 and TDP-43, but not in the IgG controls. Furthermore, there was higher co-immunoprecipitation of TDP-43 in the Tar6/Tar6 mice compared to WT mice, consistent with the overexpression of TDP-43 in the transgenic mouse line. Therefore, we observe a strong interaction between Ataxin-2 and TDP-43 was observed in both human iCell neurons and mouse brain tissue. The interaction between these two proteins was more pronounced in diseased mice compared to WT controls (Fig 1B).

### ATXN2 knock-down alleviated TDP-43 overexpression-induced neuronal loss and stress granule formation in iPSC-derived GABA neurons

Once we established that TDP-43 and Ataxin-2 interact in both iPSC neurons and in mouse brain tissues, we next hypothesized that disrupting this interaction via *ATXN2* knockdown may attenuate the toxicity induced by TDP-43. We again transduced iPSC-derived GABA neurons with AAV-TDP-43, and then transduced them with either *ATXN2*-shRNA or control-shRNA and measured *ATXN2* mRNA and protein expression via qPCR and Western blot to confirm knockdown. *ATXN2*-shRNA decreased mRNA and protein levels by greater than

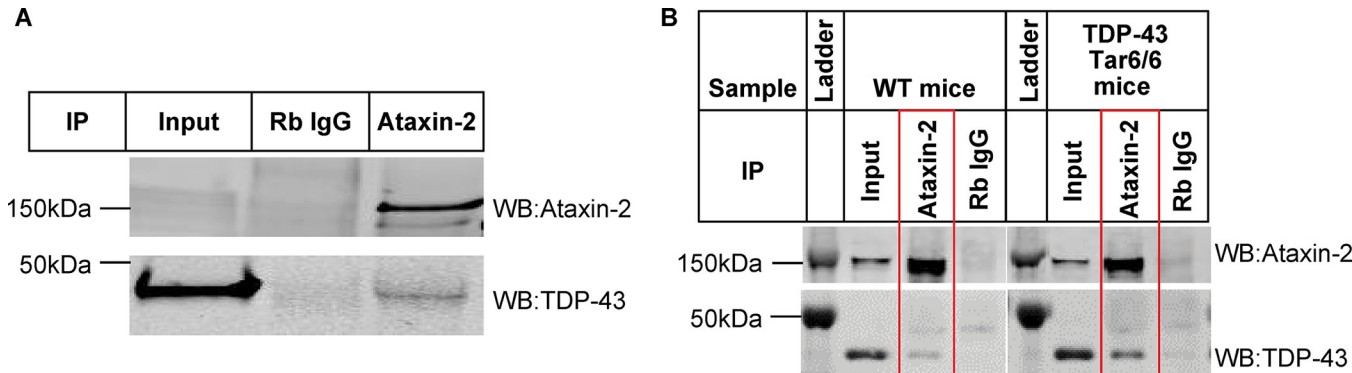

**Fig 1. Ataxin-2 co-immunoprecipitated with TDP-43 in iPSC-derived neurons and mouse brain tissue.** Ataxin-2 co-immunoprecipitated with TDP-43 in human iCell neurons (**A**) and mouse brain tissue (**B**). The iCell neurons were infected with AAV8-syn-TDP-43, and the lysate was immunoprecipitated with either IgG or Ataxin-2 antibody. The elutes were followed by Western blot with Ataxin-2 and TDP-43 antibodies. WT and Tar6/Tar6 mouse brain lysates were immunoprecipitated with IgG or Ataxin-2 antibody, respectively. The elutes were followed by Western blot with Ataxin-2 and TDP-43 antibodies.

50%, respectively compared to control-shRNA, while protein expression of TDP-43 remained unchanged (Fig 2A and 2B).

To test whether knockdown of Ataxin-2 could improve TDP-43-related toxicity, we transduced iPSC-derived GABA neurons with either AAV-TDP-43 or AAV-control to first induce toxicity through TDP-43 overexpression. AAV-TDP-43 results in significantly increased TDP-43 in the cytoplasm as well as increased cell death (Fig 2C upper panel, 2D, 2E). We then transduced the iPSC neurons with either ctrl-shRNA or ATXN2-shRNA lentivirus. Knocking down ATXN2 resulted in significantly decreased TDP-43 in the cytoplasm and improved cell survival after AAV-TDP-43 overexpression compared to the lentivirus control-shRNA (Fig 2C panel, 2D, 2E). These data suggest that knocking down ATXN2 rescues TDP-43-induced toxicity.

Ataxin-2 is known to play an important role in stress granules assembly [17]. Therefore, we asked if knocking down ATXN2 could attenuate stress granule pathology and, importantly, impact the TDP-43-containing stress granules. We again transduced iPSC GABA neurons with AAV-TDP-43 and then stressed them with 0.25mM sodium arsenite for 2 hours to induce stress granule formation. We then stained for eIFh, a stress granule marker, and TDP-43, to visualize the TDP-43-containing stress granules. We observed robust increase in the number of stress granules following sodium arsenite treatment, as well as a robust increase in the number of TDP-43-containing stress granules (Fig 2F–2H). When we co-transduced with a lentivirus expressing ATXN2-shRNA, the number of stress granules and TDP-43-containing stress granules was significantly reduced (Fig 2F–2H). Taken together, these data indicate that ATXN2-shRNA knockdown can decrease TDP-43 expression in the cytoplasm, improve neuronal survival, and rescue stress granule pathology. Ataxin-2 is known to play an important role in stress granules assembly [17]. Therefore, we asked if knocking down ATXN2 could attenuate stress granule pathology and, importantly, impact the TDP-43-containing stress granules. We again transduced iPSC GABA neurons with AAV-TDP-43 and then stressed them with 0.25mM sodium arsenite to induce stress granule formation. We then stained for eIFh, a stress granule marker, and TDP-43, to visualize the TDP-43-containing stress granules. We observed robust increase in the number of stress granules following sodium arsenite treatment, as well as a robust increase in the number of TDP-43-containing stress granules (Fig 2F–2H). When we co-transduced with a lentivirus expressing ATXN2-shRNA, the number of stress granules and TDP-43-containing stress granules was significantly reduced (Fig 2F–2H). Taken together, these data indicate that ATXN2-shRNA knockdown can decrease TDP-43 expression in the cytoplasm, improve neuronal survival, and rescue stress granule pathology.

## Ataxin-2 interacted with TDP-43 via the RRM domains and polyQ extension enhanced the interaction

After we established that Ataxin-2 and TDP-43 interact by co-immunoprecipitation, we next asked on which domain this interaction occurs in iPSC GABA neurons. It has been reported that five Phe to Leu mutations in the RRM domain abolished the ability of TDP-43 to interact with Ataxin-2 in HEK297T mammalian cells [6, 18]. To test this, we transduced human iPSC-derived iGABA neurons with either wild type scAAV8-Syn-TDP-43-flag or scAAV8-Syn-TDP-43mutRRM-flag, that have 5 Phe mutated to Leu in the RRM domains of TDP-43 as illustrated in Fig 3A, and immunoprecipitated the lysates with a flag antibody. We then performed Western blot analysis using both Ataxin-2 and TDP-43 antibodies (Fig 3B). We found significantly more Ataxin-2 in the immunoprecipitants transduced with AAV-TDP-43-flag compared to AAV-TDP-43mutRRM with even input protein levels, suggesting that the RRM mutations decreased the TDP-43-Ataxin2 interaction in iPSC GABA neurons.

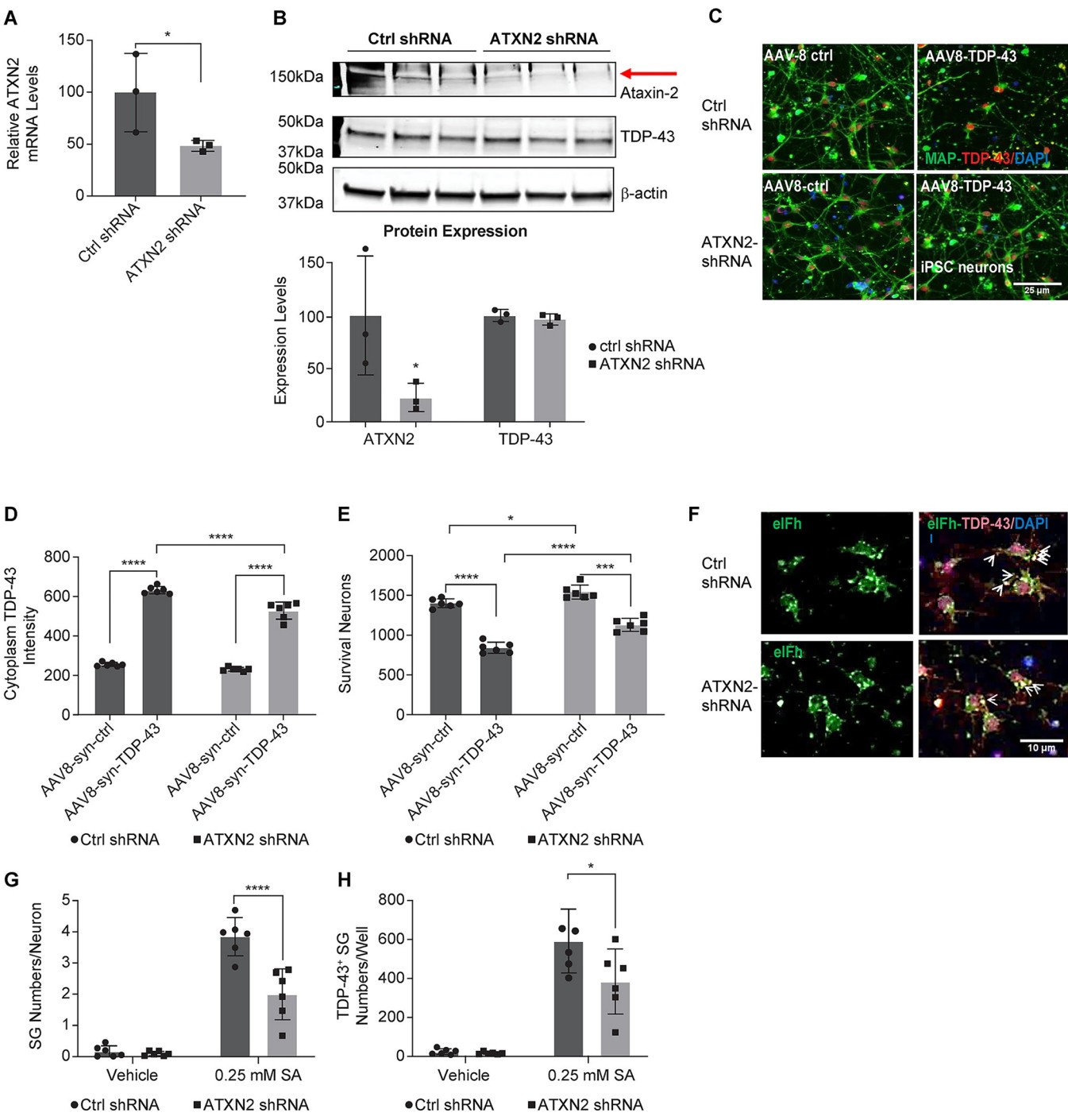

**Fig 2. *ATXN2* knock-down alleviated TDP-43 overexpression-induced neuronal loss and stress granule formation in iPSC-derived GABA neurons.**
Human iPSC-derived GABAergic neurons that were infected with LV-shRNA ctrl or LV-shRNA *ATXN2* were then transduced with AAV-control or AAV-TDP-43. **(A)** Cell lysates were analyzed by qPCR to confirm the *ATXN2* mRNA knockdown (n = 3, mean ± SD, * $p<0.05$, compared to the control group, two-sample t-test). **(B)** Western blot was performed with anti-Ataxin-2 and TDP-43 antibodies (upper panel) for protein expression levels. The images were quantified and normalized to LV-shRNA-ctrl treated cells as 100% (lower panel) (n = 3, mean ± SD, * $p<0.05$, compared to the control group, two-sample t-test). **(C)** ICC was performed with anti-TDP-43 antibody (red) and anti-Map-2 antibody (green) to visualize neuronal bodies and neurites. Nuclei were stained by Hoechst (blue). **(D)** High-content imaging quantification of cytoplasm TDP-43 (RU per cell). **(E)** Survival neurons by MAP-2 staining (per well) (n = 3, mean ± SD * $p<0.05$, ** $p<0.01$, *** $p<0.001$, **** $p<0.0001$ two-way ANOVA followed by Šidák's multiple comparisons test). **(F)** High-content imaging of ICC with anti-TDP-43 antibody (red) and anti-eIFh antibody (green) to visualize stress granules. Nuclei were stained by Hoechst (blue). The iPSC-derived GABA neurons that were infected with LV-shRNA-ctrl and LV-shRNA-ATXN2 then infected with AAV-TDP-43 before treated with Sodium Arsenite (SA) for 2hrs to induce stress granule formation. Arrows indicate stress granules containing TDP43. **(G)** High-content imaging quantification of eIFh+ stress

granule numbers per neuron (n = 9/group, 2 independent experiments, mean ± SD, ***$p<0.0001$, two-way ANOVA followed by Šidák's multiple comparisons test). **(H)** High-content imaging quantification of TDP-43 positive stress granule numbers per well, which is defined as stress granules that have a well-defined TDP-43 staining overlap (n = 9/group, 2 independent experiments, mean ± SD, ****$p<0.0001$, two-way ANOVA followed by Šidák's multiple comparisons test).

To further explore the role of the RRM mutation in the TDP-43-Ataxin2 interaction using another methodology, we transfected mammalian constructs expressing Ataxin-2 with either TDP-43-flag or TDP-43mutRRM-flag in HEK293 cells. We then used an AlphaLISA proximity assay to measure the interaction between Ataxin-2 (recognized by biotinylated Ataxin-2 antibody/streptavidin donor beads) and the flag-tagged TDP-43 construct (recognized by flag-acceptor beads). We observed a 22-fold increase in the interaction of WT TDP-43 and Ataxin-2, compared with only a 2-fold increase in the TDP-43-Ataxin-2 interaction when the TDP-43 had the RRM mutation (Fig 3C). We then tested whether we could increase this interaction by increasing the poly-Q region of Ataxin-2. We transfected cells with either WT ATXN2, ATXN2-31Q, or ATXN2-39Q and found increasing interaction with increase poly-Q region as measured by AlphaLISA (Fig 3D). These results further confirmed that the TDP-43-Ataxin-2 interaction was significantly reduced with the RRM mutations.

We next sought to test if the TDP-43-Ataxin2 interaction can be enhanced with polyQ expansion. To do this, we again transfected HEK293 cells with TDP-43-flag and either ATXN2, ATXN2-31Q or ATXN2-39Q and performed the AlphaLISA proximity assay to quantitatively measure the interaction between Ataxin-2 and the flag-tagged TDP-43. We found a stepwise increase in the TDP-43-Ataxin-2 binding, with increasing polyQ repeats enhancing this interaction. These data further support that Ataxin-2 interacts with TDP-43 and this interaction may play an important role in TDP-43 toxicity and in disease manifestation with polyQ expansions.

## TDP-43 RRM mutation attenuated TDP-43-induced toxicity in iPSC-derived neurons

Since our data suggests that the RRM domain regulates the interaction between TDP-43 and Ataxin-2 in iPSC neurons, we hypothesized that disrupting this interaction may attenuate TDP-43 related toxicity similarly to the ATXN2 knockdown. Therefore, we transduced iPSC GABA neurons with either AAV-control, AAV-TDP-43, AAV-TDP-43mutRRM for 14 days and used high content imaging to quantify neuronal survival (number of MAP2+ neurons), neuronal death (percent of cells with cleaved caspase3), or levels of cytoplasmic TDP-43. Compared to AAV-control, overexpression of WT TDP-43 resulted in significantly fewer MAP2 + neurons, increased cleaved caspase3, and increased cytoplasmic TDP-43, indicating significant cell death from TDP-43 toxicity (Fig 4A–4D). However, when TDP-43 was overexpressed with the RRM mutations, the loss of MAP2+ neurons and percent of neurons with cleaved caspase3 was completely rescued back to levels similar to AAV-control (Fig 4B and 4C). AAV-TDP-43mutRRM also results in significantly less TDP-43 in the cytoplasm compared to WT TDP-43 overexpression (Fig 4D). Therefore, the RRM domain plays an important role in TDP-43- Ataxin-2 interaction and TDP-43 mediated toxicity.

Since we established that disrupting the TDP-43-Ataxin-2 interaction either via knockdown of TDP-43 or by mutating the TDP-43 RRM domain rescued TDP-43 pathology and improved cell survival, we next aimed to identify other potential therapeutic targets that can influence the interaction of these proteins. Therefore, we used co-immunoprecipitation followed by LC-MS analysis to identify the Ataxin2-TDP-43 interactome. The goal of this experiment was to identify binding co-factors that may block or interrupt this interactome that we could target

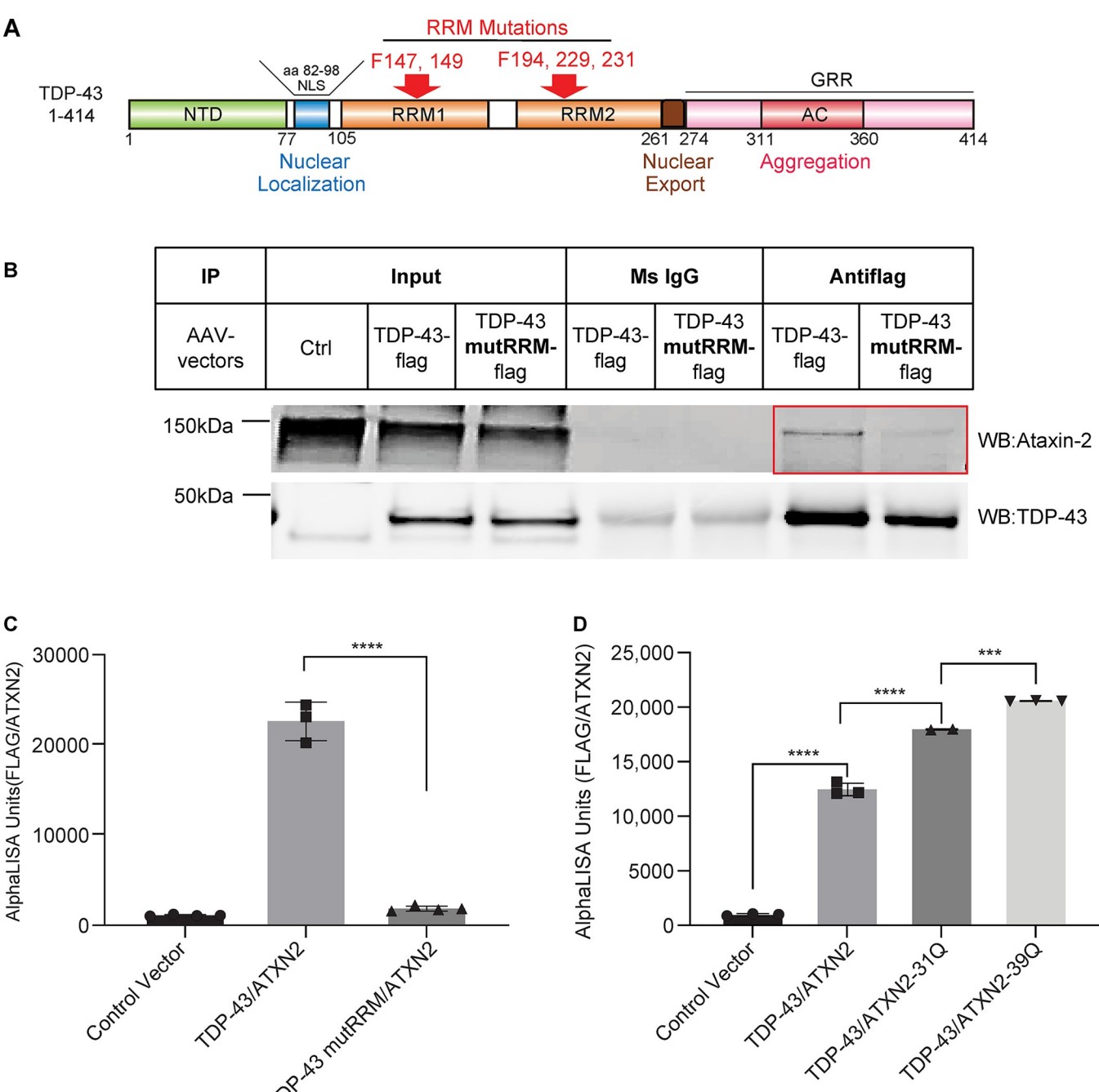

**Fig 3. Ataxin-2 interacted with TDP-43 via the RRM domains and polyQ extension enhanced the interaction. (A)** Schematic representation of TDP-43 protein and the RRM mutations that abolish its interaction with RNA binding proteins. **(B)** Co-immunoprecipitation and western blot of TDP-43 and TDP-43 RRM mutant with Ataxin-2. Human iPSC-derived GABA neurons were infected with scAAV8-Syn-TDP-43-flag or scAAV8-Syn-TDP-43mutRRM-flag, and the lysates were immunoprecipitated with Flag antibody. Western blot was performed with Ataxin-2 and TDP-43 antibodies. **(C)** Proximity AlphaLISA for the interaction between TDP-43 and TDP-43mutRRM with Ataxin-2. HEK293 cells were transfected with either empty vector, or vectors co-overexpressing ATXN2 and TDP-43-flag or ATXN2 and TDP-43mutRRM-flag, respectively. Cell lysates were analyzed by proximity AlphaLISA (n = 3, mean ± SD * $p<0.05$, ** $p<0.01$, *** $p<0.001$, **** $p<0.0001$ one-way ANOVA followed by Tukey's multiple comparison test). **(D)** Proximity AlphaLISA for the interaction between either Ataxin-2, or Ataxin-2-31Q, or Ataxin-2-39Q and TDP-43. HEK293 cells were transfected with either empty vector, or vectors overexpressing TDP-43-flag and ATXN2, TDP-43-flag and ATXN2-31Q or TDP-43-flag and ATXN2-39Q, respectively. Cell lysates were analyzed by proximity AlphaLISA (n = 3, mean ± SD * $p<0.05$, ** $p<0.01$, *** $p<0.001$, **** $p<0.0001$ one-way ANOVA followed by Tukey's multiple comparison test).

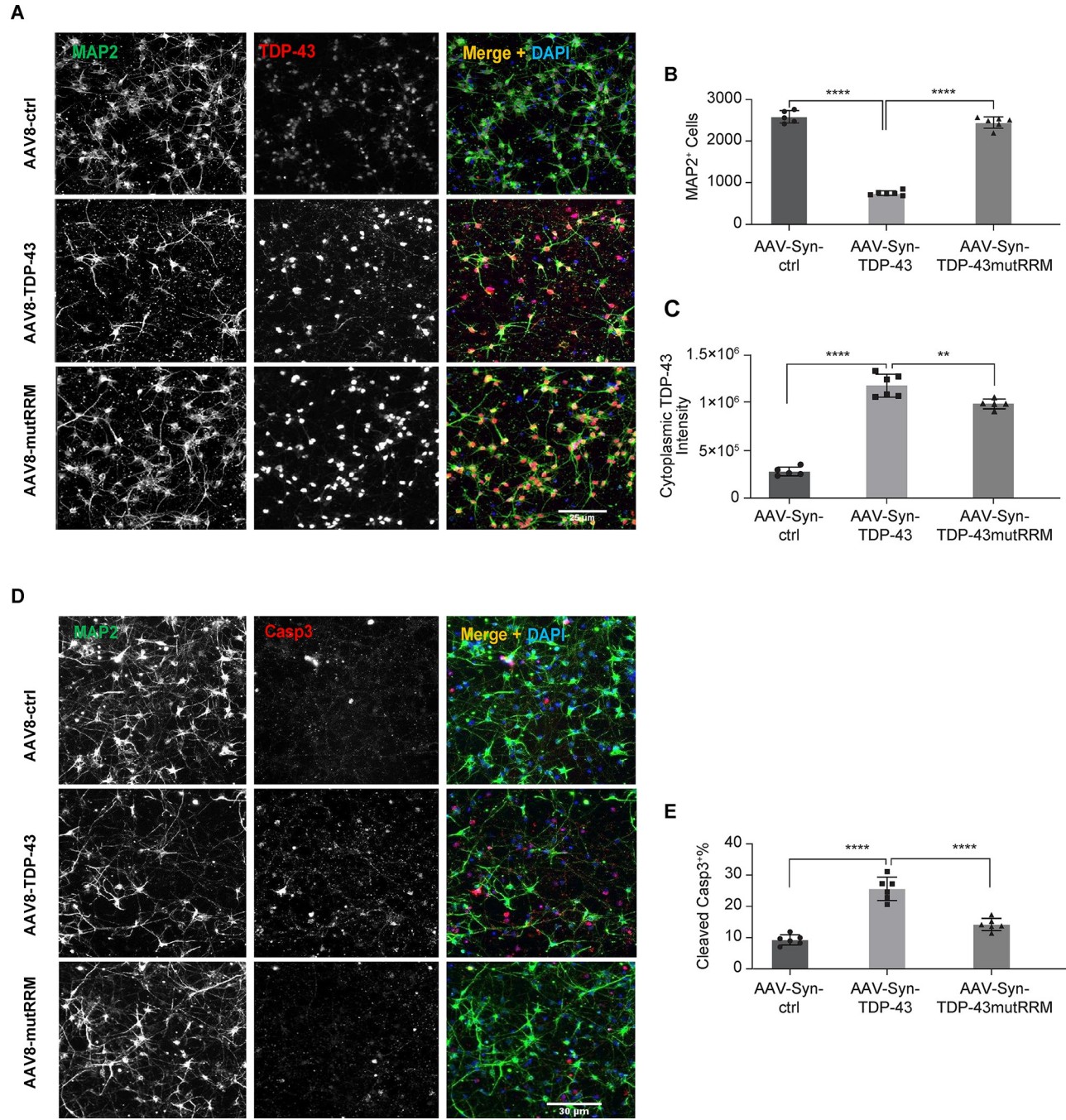

**Fig 4. TDP-43 RRM mutant abolished TDP-43-induced toxicity in iPSC-derived neurons.** (A) ICC of human iPSC-derived GABA neurons that were infected with AAV-control (upper panel), AAV-TDP-43 (middle panel) or AAV-TDP-43mutRRM (lower panel). Immunostaining was performed with anti-TDP-43 antibody (red) and anti-Map-2 antibody (green) to visualize neuronal bodies and neurites. Nuclei were stained by Hoechst (blue). (B) High-content imaging quantification of MAP-2 positive cells per well (n = 3, mean ± SD * $p<0.05$, ** $p<0.01$, *** $p<0.001$, **** $p<0.0001$ one-way ANOVA followed by Tukey's multiple comparison test). (C) High-content imaging quantification Caspase 3 positive cells as percentage to total cells per well (n = 3, mean ± SD * $p<0.05$, ** $p<0.01$, *** $p<0.001$, **** $p<0.0001$ one-way ANOVA followed by Tukey's multiple comparison test). (D) High-content imaging quantification of cytoplasm TDP-43 intensity per well of RU (n = 3, mean ± SD * $p<0.05$, ** $p<0.01$, *** $p<0.001$, **** $p<0.0001$ one-way ANOVA followed by Tukey's multiple comparison test).Co-Immunoprecipitation followed by Mass Spectrometry (IP-MS) identified Ataxin-2 interactome in iPSC derived neurons.

in future experiments. We transduced iPSC neurons with either AAV-control or AAV-TDP-43 and co-immunoprecipitated the lysate with either Ataxin-2 or IgG isotype control antibodies. After tryptic digestion, we performed LC-MS followed by SWATH data analysis to identify binding co-factors in both the TDP-43 overexpression and control conditions (Fig 5A). Both the TDP-43 and normal control groups were compared to their own IgG controls, and so comparisons in this figure are made between the ATXN2 IP pulldown and the IgG isotype control pulldown. We first performed a Western blot analysis to confirm TDP-43 overexpression and the endogenous expression of Ataxin-2 (Fig 5B). Based on our quantitative LC-MS analysis, we identified a total of 426 proteins that interact with the ATXN-2 IP in the TDP-43 overexpressed neurons and 421 proteins in the normal neurons (endogenous TDP-43 level), respectively (S1 Table) as shown in the volcano plots (Fig 5C and 5D). Proteins that were found to interact in the control neurons (normal) were endogenous interactors, and proteins that interact with the TDP-43 overexpression neurons (TDP-43) were possible disease-state interactors. Some of the proteins with the fold change >3 for the Mean Average of Ataxin-2 (n = 3) over Mean Average IgG (n = 3) and the adjusted p-value<0.05 were highlighted in red (Fig 5C, 5D). Of note, the most highly significant protein in the TDP-43 overexpression condition was TAF15. We then performed Pathway Analysis (IPA) with the identified proteins found to interact with TDP-43 and Ataxin-2 in both the disease and endogenous state in order to map out the significant canonical pathways, biological functions, and disease pathways these proteins involved (Fig 5E). Pathways involved in RNA processes, neurological disorders, inflammation, and basic cellular processes were among the top hits. As pathways involving progressive neurological disorders and RNA processing are of great interest to ALS therapy, the network of these two pathways were algorithmically generated (Fig 5F). In the future, we or others may target some of the proteins identified in these pathways, particularly those that were predicted to either activate or inactivate the pathway.

We then analyzed the fold-change comparison between the TDP-43 overexpression versus normal conditions (S2 Table). Proteins with the fold change >1.5 or -1.5 for TDP-43 (n = 3) over Normal (n = 3) were highlighted in red (Fig 6A) in the plot indicating either up- or down-regulated proteins in the condition of TDP-43 overexpression. Proteins greater than 1.5 fold change such as TARDBP, RAN, and MTA2 are upregulated, and proteins greater than –1.5 fold change such as TAF15 and others highlighted in red are downregulated compared to controls. In addition, we created a Venn diagram comparing the unique and common proteins between the two conditions and found that TDP-43 and normal conditions had 403 proteins in common, while the TDP-43 group had 23 unique proteins and the normal group had 18 unique proteins (Fig 6B). TAF15, while very highly significantly upregulated in the TDP43 overexpression group, is also expressed in the normal condition, and so therefore appears in the overlap category. Of these, we hypothesized that the proteins differentially regulated in the disease-state condition may be of the most interest to potentially target to disrupt the Ataxin2-TDP-43 interaction.

## Discussion

ALS is a fatal neurodegenerative disease, and most patients live only 3–5 years following diagnosis [19, 20]. Current approved therapies (riluzole and edaravone) provide only moderate symptom relief, and so there is a critical need to develop new therapies that can modify the disease [20]. Riluzole is thought to decrease glutamatergic excitotoxicity and can extend patient survival by about 3 months [19]. Edaravone is a reactive oxygen species (ROS) scavenger and may slow disease progression and decline on the ALSFRS-R during early stages of the disease,

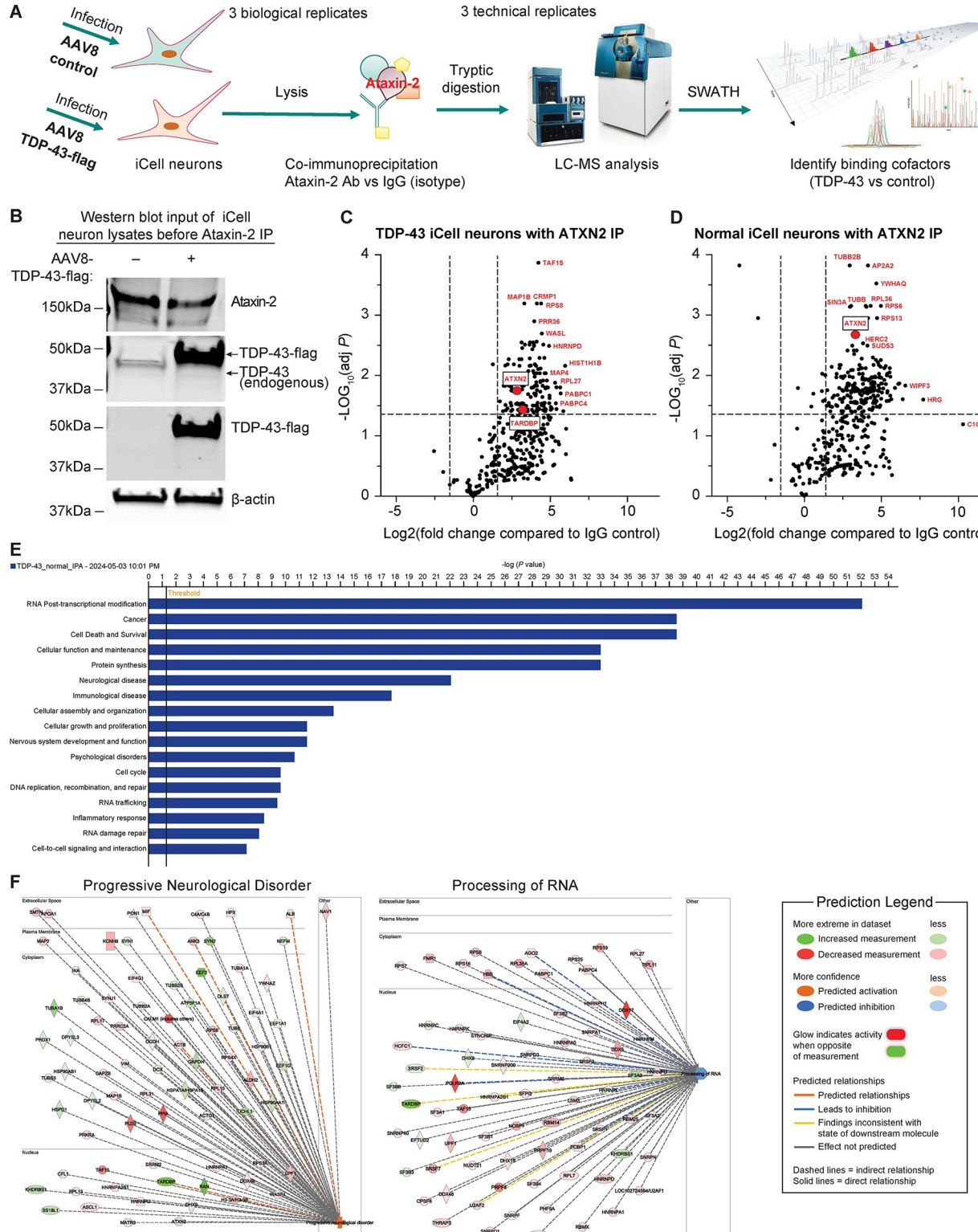

**Fig 5. IP-MS identified ATXN2 interactome in the conditions of endogenous (Normal) or TDP-43 overexpression (TDP-43) in iPSC-derived GABA neurons. (A)** Schematic representation of the IP-MS procedure. Human iPSC-derived GABA neurons were infected with either AAV-ctrl or AAV-TDP-43, and the lysate was immunoprecipitated with either isotype IgG control or Ataxin-2 antibody with 3 biological replicates. The elutes were followed by tryptic digestion and LC-SWATH MS analysis with 3 technical replicates to identify binding cofactors of TDP-43 overexpression vs control (endogenous TDP-43). **(B)** Western blot of the input cell lysates before Ataxin-2 IP. Aliquots of the cell lysates

of either AAV-ctrl or AAV-TDP-43 infected neurons were analyzed by Western blot with Ataxin-2 and TDP-43 antibodies. **(C, D)** Volcano plot of all interacting proteins identified from TDP-43 overexpression (TDP-43) vs endogenous expression (Normal). They were shown with Fold Change = Mean Ave of Ataxin-2 (3 bioreplicates) over Mean Ave IgG (3 bioreplicates). Horizontal dotted line for Adj. p-value = 0.05, Welch T-Test (assume unequal variances), and vertical dotted lines for Fold Change = 3 or -3. **(E)** IPA analysis of the canonical pathways, biological functions, and disease pathways that the hits involved. The line bar represents the threshold of significance (p = 0.05). **(F)** The interaction networks of the hits functionally related to progressive neurological disorders and progressing of RNA.

but with limited efficacy [19, 21–24]. Because of the limited efficacy of the currently approved drugs for ALS, there is a huge unmet medical need for disease modifying therapies.

In recent years, advances in RNA therapeutics have led to more clinical trials for ASO therapy for ALS [25]. ASOs can modulate gene expression and protein levels, without altering the host genome [25]. ASOs may be a good treatment option for patients with a confirmed disease mutation, such as tofersen (approved by the FDA in 2023) for patients with an SOD1 mutation [26]. ASO therapy for ATXN2 is currently being tested in clinical trial (ClinicalTrials.gov: NCT04494256). Despite some promise, there are some challenges associated with ASO therapies including blood brain barrier penetrance and the possibility for immune activation. Because ASOs do not cross the blood brain barrier, delivery is invasive and requires intrathecal or intracerebroventricular injection directly to the CSF [25]. Therefore, pursuing small molecules as potential therapies still may hold promise. We chose to explore a unique approach to find potential druggable targets that interact with Ataxin-2 and TDP-43, with the thought that in the future, these targets might be appropriate for small molecules.

In ALS, TDP-43 is mislocalized to the cytoplasm, resulting in both a loss of function in the nucleus and a gain of toxic function in the cytoplasm [27, 28]. We chose to use an iPSC model which overexpresses TDP-43 via an AAV vector because ATXN2 has been shown to be an important modifier of TDP-43 aggregation and gain of function toxicity [8, 9]. Additionally, we have previously reported this model to recapitulate TDP-43-induced toxicity, including decreased neurite length and decreased cell survival [14]. While this is a useful model to study TDP-43 toxicity and gain of function in the cytoplasm, we would not necessarily expect to see a decrease of TDP-43 in the nucleus. Other papers studying other TDP-43-related cryptic exons, such as *STMN2*, *UNC13A*, and others, often use TDP-43 si-RNA cell models to knockdown levels of TDP-43 and mimic a loss of function phenotype [29–31]. In future experiments, we may explore differences in the ATXN2-TDP-43 interaction in loss of function versus gain of function models of disease. Additionally, the RRM domain of the TDP-43 protein notably binds RNA, and therefore, it is a possibility that the Ataxin-2 and TDP-43 interaction may also be mediated through RNA. Future studies may seek to explore the contribution of RNA to this interaction.

This study identified 403 proteins that interact with TDP-43 and Ataxin-2 in both normal and disease states. These proteins are involved in pathways such as neurological disease, RNA processing, immune/inflammatory response, and basic cell processes such as cell death and survival, cell growth, cell signaling, etc., indicating that proteins that bind to both TDP-43 and Ataxin-2 have important and varied roles in the cell. We explored in more detail the proteins involved in neurological disease and RNA processing pathways because of TDP-43's role as an RNA binding protein and its implication in ALS. Several of the proteins in these pathways were tubulin, neurofilament, or cytoskeletal related proteins, which could be of interest as other tubulins and cytoskeletal proteins have been shown to have mutations that directly cause ALS [32–34]. Synaptic proteins such as SYN1 and SYN2 may also be of interest in future research as synaptic dysfunction is widely studied in ALS [35, 36]. In addition, proteins involved in the inflammatory pathways, which we did not further explore, may be of interest for future research because of the well-known role of neuroinflammation in ALS [37]. Since

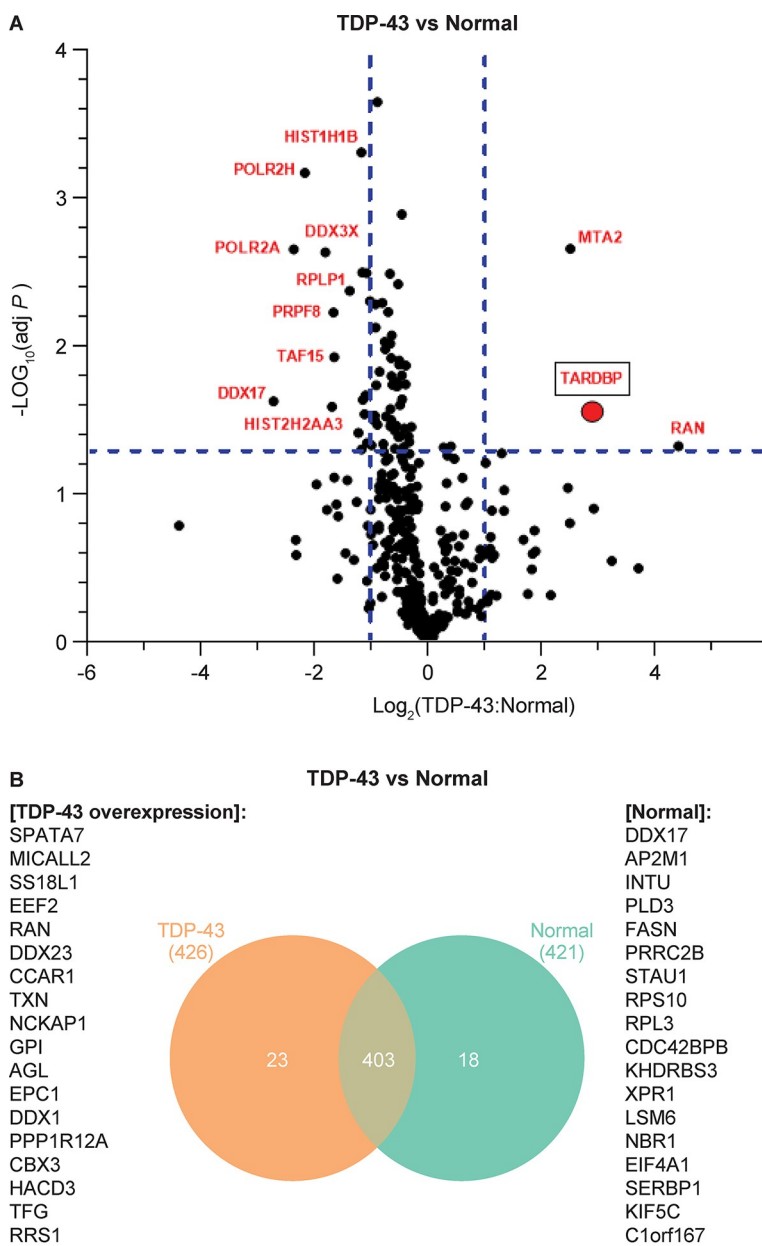

**Fig 6. Interactome hits of TDP-43 overexpression vs normal. (A)** Volcano plot of hit overlay of fold changes for TDP-43 overexpression over Normal (endogenous). Proteins highlighted in red have a fold change>2 compared to normal; Welch T-Test (assume unequal variances), Adj. p-value<0.05. Horizontal dotted line for Adj. p-value = 0.05, Welch T-Test (assume unequal variances), and vertical dotted lines for Fold Change = 2 or -2. **(B)** Venn plot summary of hits distinct to TDP-43 (overexpression) vs Normal (endogenous).

our analysis has shown many different pathways known to be implicated in ALS, there is potential that some of our proteins that were hits in the interactome analysis could be potential therapeutic targets in the future.

In addition to the 403 proteins that interact in both disease and endogenous states, we also found 23 unique proteins that interact with TDP-43 and Ataxin-2 under TDP-43 overexpression conditions. We hypothesize that the proteins upregulated only in the disease state may also be potential therapeutic targets, and by modulating these proteins either genetically or with a small molecule, we may be able to attenuate TDP-43-related toxicity. Among the proteins we found that were implicated in both the neurological disease and RNA processing pathway using IPA was TAF15 (Fig 5F). We hypothesize that these results suggest that the Ataxin-2 interactome may play an important role in TDP-43 mediated toxicity, but more research is needed to confirm this finding in the future. For example, future research may seek to overexpress *TAF15*, screen for *TAF15* small molecule activators or inhibitors, or explore the interaction between *TAF15* and TDP-43 to test if *TAF15* activity can rescue TDP-43 toxicity.

In summary, we show in this paper that TDP-43 and Ataxin-2 interact via the RRM domain, and interfering with this interaction either by knocking down ATXN2 or by mutating the RRM domain can attenuate TDP-43 related phenotypes. We then performed co-immuno-precipitation followed by mass spectrometry to identify proteins in the TDP-43-Ataxin-2 interactome in both normal and TDP-43 overexpression conditions. One of these identified proteins, TAF15, has been previously implicated in ALS. This study comprehensively characterizes the Ataxin-2 and TDP-43 interactome, which can be used to identify potential therapeutic targets in future work.

## Supporting information

**S1 Table. LC-MS identified total proteins that interact with the ATXN-2 IP in the TDP-43 overexpressed neurons vs in the normal neurons.** Human iPSC-derived GABA neurons were infected with either AAV-ctrl or AAV-TDP-43, and the lysate wasimmunoprecipitated with either isotype IgG control or Ataxin-2 antibody with 3 biological replicates. The elutes were followed by tryptic digestion and LC-SWATH MS analysis with 3 technical replicates to identify the total binding cofactors of TDP-43 overexpression vs normal.
(XLSX)

**S2 Table. LC-MS identified protein fold-change comparison between TDP-43 overexpression versus normal neurons.** Human iPSC-derived GABA neurons were infected with either AAV-ctrl or AAV-TDP-43, and the lysate wasimmunoprecipitated with either isotype IgG control or Ataxin-2 antibody with 3 biological replicates. The elutes were followed by tryptic digestion and LC-SWATH MS analysis with 3 technical replicates. The fold change of proteins between Ataxin-2 antibody IP vs IgG control IP was analyzed. In addition, the fold change of TDP-43 overexpression versus normal conditions was analyzed.
(XLSX)

**S1 File. Original uncropped Western blot images and additional method description for IP-MS.**
(DOCX)

## Acknowledgments

We thank for Xiaohai Wang and Jon Sugam for reviewing and commenting the manuscript, and Ann Marie Norris for technical editing and formatting.

## Author Contributions

**Conceptualization:** Yuan Tian, Aaron Zefrin Fernandis, Sophie Parmentier-Batteur, Jason M. Uslaner, Sean M. Smith.

**Data curation:** Yuan Tian, Nicolette Heinsinger, Yinghui Hu, U-Ming Lim, Yi Wang.

**Formal analysis:** Yuan Tian, Yinghui Hu, U-Ming Lim, Yi Wang.

**Investigation:** Yuan Tian.

**Methodology:** Nicolette Heinsinger, Yinghui Hu, U-Ming Lim, Yi Wang, Aaron Zefrin Fernandis, Sophie Parmentier-Batteur.

**Resources:** Sophie Parmentier-Batteur, Jason M. Uslaner, Sean M. Smith.

**Supervision:** Yuan Tian, Aaron Zefrin Fernandis, Becky Klein.

**Writing – original draft:** Yuan Tian, Nicolette Heinsinger.

**Writing – review & editing:** Yuan Tian, Nicolette Heinsinger, Jason M. Uslaner, Sean M. Smith.

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
