## [Decision Letter · Decision Letter 0]

11 Nov 2024

PONE-D-24-30547Deciphering the interactome of Ataxin-2 and TDP-43 in iPSC-derived neurons for potential ALS targetsPLOS ONE

Dear Dr. Tian,

Thank you for submitting your manuscript to PLOS ONE. After careful consideration, we feel that it has merit but does not fully meet PLOS ONE’s publication criteria as it currently stands. Therefore, we invite you to submit a revised version of the manuscript that addresses the points raised during the review process.

We look forward to receiving your revised manuscript.

Kind regards,

Asif Ali

Academic Editor

PLOS ONE

Journal Requirements:

3. Thank you for stating the following in the Competing Interests section: “All authors were employees of Merck & Co., Inc., Rahway, NJ, USA at the time of this study. Employment does not alter authors' adherence to the journal's policies on conflicts of interest or sharing data and materials.”

We note that one or more of the authors are employed by a commercial company: Merck & Co., Inc., Rahway, NJ, USA

1. Please provide an amended Funding Statement declaring this commercial affiliation, as well as a statement regarding the Role of Funders in your study. If the funding organization did not play a role in the study design, data collection and analysis, decision to publish, or preparation of the manuscript and only provided financial support in the form of authors' salaries and/or research materials, please review your statements relating to the author contributions, and ensure you have specifically and accurately indicated the role(s) that these authors had in your study. You can update author roles in the Author Contributions section of the online submission form. Please also include the following statement within your amended Funding Statement. “The funder provided support in the form of salaries for authors [insert relevant initials], but did not have any additional role in the study design, data collection and analysis, decision to publish, or preparation of the manuscript. The specific roles of these authors are articulated in the ‘author contributions’ section.” If your commercial affiliation did play a role in your study, please state and explain this role within your updated Funding Statement. 2. Please also provide an updated Competing Interests Statement declaring this commercial affiliation along with any other relevant declarations relating to employment, consultancy, patents, products in development, or marketed products, etc. Within your Competing Interests Statement, please confirm that this commercial affiliation does not alter your adherence to all PLOS ONE policies on sharing data and materials by including the following statement: "This does not alter our adherence to PLOS ONE policies on sharing data and materials.” (as detailed online in our guide for authors http://journals.plos.org/plosone/s/competing-interests) . If this adherence statement is not accurate and there are restrictions on sharing of data and/or materials, please state these. Please note that we cannot proceed with consideration of your article until this information has been declared. Please include both an updated Funding Statement and Competing Interests Statement in your cover letter. We will change the online submission form on your behalf.

6. PLOS ONE now requires that authors provide the original uncropped and unadjusted images underlying all blot or gel results reported in a submission’s figures or Supporting Information files. This policy and the journal’s other requirements for blot/gel reporting and figure preparation are described in detail at https://journals.plos.org/plosone/s/figures#loc-blot-and-gel-reporting-requirements and https://journals.plos.org/plosone/s/figures#loc-preparing-figures-from-image-files. When you submit your revised manuscript, please ensure that your figures adhere fully to these guidelines and provide the original underlying images for all blot or gel data reported in your submission. See the following link for instructions on providing the original image data: https://journals.plos.org/plosone/s/figures#loc-original-images-for-blots-and-gels. In your cover letter, please note whether your blot/gel image data are in Supporting Information or posted at a public data repository, provide the repository URL if relevant, and provide specific details as to which raw blot/gel images, if any, are not available. Email us at plosone@plos.org if you have any questions.

Additional Editor Comments:

 The manuscript by Tian et al. offers a comprehensive investigation into the interaction between Ataxin-2 and TDP-43, two key proteins implicated in Amyotrophic Lateral Sclerosis (ALS). Using iPSC-derived GABA neurons, the authors convincingly demonstrate that the interaction between Ataxin-2 and TDP-43 is mediated by the RNA Recognition Motif (RRM) of Ataxin-2. They further highlight the role of Ataxin-2 in stress granule dynamics and its influence on TDP-43-associated neurotoxicity. However, Reviewer #1 notes the need to address the possibility of RNA-mediated interactions and suggests RNase treatment as a control. Additionally, Figure 7's relevance is questioned due to the minimal effect observed with TAF-15 knockdown. Minor clarifications, such as defining TARDBP, detailing stress duration, and improving figure clarity, are also recommended. Reviewer #2 finds the study impactful and suggests including raw data and analysis details for enhanced transparency. Overall, this work provides new insights into ALS pathology and lays the groundwork for potential therapeutic targets.

Reviewers' comments:

Reviewer's Responses to Questions

**Comments to the Author**

1. Is the manuscript technically sound, and do the data support the conclusions?

Reviewer #1: Yes

Reviewer #2: Yes

2. Has the statistical analysis been performed appropriately and rigorously? 

Reviewer #1: Yes

Reviewer #2: Yes

3. Have the authors made all data underlying the findings in their manuscript fully available?

Reviewer #1: Yes

Reviewer #2: Yes

4. Is the manuscript presented in an intelligible fashion and written in standard English?

Reviewer #1: Yes

Reviewer #2: Yes

5. Review Comments to the Author

Reviewer #1: The paper investigates the interaction between Ataxin-2 and TDP-43, two proteins implicated in Amyotrophic Lateral Sclerosis (ALS). Primarily using human iPSC-derived GABA neurons, the authors show that Ataxin-2 and TDP-43 interact in cells and that this interaction is dependent on the RRM of Ataxin-2. They also show that Ataxin-2 depletion decrease stress granule formation and decreases neuron toxicity associated with TDP-43 overexpression. Finally, they perform Ataxin-2 IP-MS in the neuronal culture to identify interaction partners. Overall, the paper provides a comprehensive analysis of the Ataxin-2 and TDP-43 interactome, offering new insights into the molecular mechanisms underlying ALS and identifying potential targets for future therapeutic development. The experiments are carefully performed and well described.

Major Comments:

1. The authors clearly demonstrate that mutations in the RRM of TDP-43 reduce the pulldown of Ataxin-2, however they do not show that this interaction is direct. As the RRM of TDP-43 presumably binds RNA, this interaction may be mediated through RNA. The authors should either make this caveat clear, or even better, try treating their lysates with RNase prior to pulldown to rule out an RNA-mediated interaction.

2. I do not think that Figure 7 adds much scientifically. The TAF-15 knockdown does lead to slight neuronal death, but this effect is not amplified in the context of TDP-43 overexpression. Thus, there is no evidence that TAF15 is modulating TDP-43 mediated toxicity as the authors suggest it may be.

Minor Comments:

340: Clarify that TARDBP encodes TDP-43

375: How long was the stress? Also mention in figure legend

Figure 2G: please make clear what the SG marker is

Figure 3B: Do you think this interaction is mediated by RNA (RRM being the RNA-binding domain). Can you do an RNase treatment before the IP?

404: HEK293T?

Fig. 5C/D: label X-axis directly with FC compared to IgG control

Fig. 7B: Coloring is confusing (looks the same as the AAV-syn-TDP-43)

Reviewer #2: The manuscript titled "Deciphering the Interactome of Ataxin-2 and TDP-43 in iPSC-Derived Neurons for Potential ALS Targets" by Tian et al. presents a valuable study on the interactome of TDP-43 and Ataxin-2. This research is of considerable interest to those studying ALS protein characterization, neurobiology, and related fields. I anticipate that this article will be widely cited.

The manuscript is well-written, and I commend the authors for their efforts. I have only one minor suggestion for improvement: I recommend that the authors provide all raw data in a supplementary file along with a detailed description of their data analysis process, as this would be beneficial for readers.

6. PLOS authors have the option to publish the peer review history of their article (what does this mean?). If published, this will include your full peer review and any attached files.

Reviewer #1: No

Reviewer #2: **Yes: **Pawan Kumar

---

## [Author Response · Author response to Decision Letter 0]

27 Nov 2024

November 25th, 2024

Re: PLOS ONE Decision: Revision required [PONE-D-24-30547] - [EMID:43544307ce9c1c20]

Asif Ali

Academic Editor

PLOS ONE

Dear Editor Asif,

Thank you for considering the publication of our manuscript "Deciphering the interactome of Ataxin-2 and TDP-43 in iPSC-derived neurons for potential ALS targets". The authors thank the reviewers for their enthusiastic response to our work and insightful comments for improvement. We are excited that the reviewers feel our work will be “of considerable interest” to the field, that “our experiments are carefully performed and well described”, and they “anticipate that this article will be widely cited”. We addressed the reviewers’ comments and are excited to present this revised and significantly strengthened manuscript for you to review. Below you will find an itemized list of the reviewers' comments and our response describing revisions made.

The authors’ responses are in blue, and changes made in the manuscript are labeled with page and line numbers and in blue italic.

Reviewer #1: The paper investigates the interaction between Ataxin-2 and TDP-43, two proteins implicated in Amyotrophic Lateral Sclerosis (ALS). Primarily using human iPSC-derived GABA neurons, the authors show that Ataxin-2 and TDP-43 interact in cells and that this interaction is dependent on the RRM of Ataxin-2. They also show that Ataxin-2 depletion decrease stress granule formation and decreases neuron toxicity associated with TDP-43 overexpression. Finally, they perform Ataxin-2 IP-MS in the neuronal culture to identify interaction partners. Overall, the paper provides a comprehensive analysis of the Ataxin-2 and TDP-43 interactome, offering new insights into the molecular mechanisms underlying ALS and identifying potential targets for future therapeutic development. The experiments are carefully performed and well described.

Major Comments:

 1. The authors clearly demonstrate that mutations in the RRM of TDP-43 reduce the pulldown of Ataxin-2, however they do not show that this interaction is direct. As the RRM of TDP-43 presumably binds RNA, this interaction may be mediated through RNA. The authors should either make this caveat clear, or even better, try treating their lysates with RNase prior to pulldown to rule out an RNA-mediated interaction.

We agree with the reviewer that perhaps this interaction is mediated by RNA, and we carefully considered their suggestion. However, the IP methodology that we used here may not be sensitive enough to pick up on small changes that may result from using RNase treatment as the reviewer suggests. We acknowledge the possibility that using RNase treatment before IP may cause the protein-protein interactions to be weaker; however, we feel that this experiment would require more sensitive methodology and much further experimentation. This experiment would be very interesting for follow up studies in the future.

Additionally, we added the following text to the discussion on line 656 to make this caveat clear as the reviewer suggests: “Additionally, the RRM domain of the TDP-43 protein notably binds RNA, and therefore, it is a possibility that the Ataxin-2 and TDP-43 interaction may also be mediated through RNA. Future studies may seek to explore the contribution of RNA to this interaction.”

 2. I do not think that Figure 7 adds much scientifically. The TAF-15 knockdown does lead to slight neuronal death, but this effect is not amplified in the context of TDP-43 overexpression. Thus, there is no evidence that TAF15 is modulating TDP-43 mediated toxicity as the authors suggest it may be.

We agree with the reviewer that the TAF15 story needs more experimentation that would be out of scope of this manuscript. Therefore, we decided to remove Figure 7 from the paper because we agree that as it stands alone now is a bit confusing to the reader. However, we did leave some speculation about the effect of TAF 15 in our discussion because it was an interesting finding from our Mass Spectrometry analysis that has been supported by recent literature, and we wanted to highlight its interest to the field.

Minor Comments:

 340: Clarify that TARDBP encodes TDP-43

We added this clarification on line 347. The texts now reads: Tar6/Tar6 is a transgenic mouse line that overexpresses human wild-type TARDBP (encodes for TDP43) at about 1.2-fold over endogenous protein level(16).

 375: How long was the stress? Also mention in figure legend

We added that the stress was for 2 hours to both the results on line 413 and figure legend on line 389.

 Figure 2G: please make clear what the SG marker is

We added this clarification to the 2G figure legend. It now reads: High-content imaging quantification of eIFh+ stress granule numbers per neuron...

 Figure 3B: Do you think this interaction is mediated by RNA (RRM being the RNA-binding domain). Can you do an RNase treatment before the IP?

We agree with the reviewer that it is possible that this interaction may be mediated by RNA, though fully exploring this possibility would require more sensitive methodology that is outside the scope of this paper. Please see our comments above for a more complete explanation. 

 404: HEK293T?

HEK293T cells are a derivative of HEK293 cells, that have been transformed with the SV40 large T antigen. This mutation results in faster growth and higher transfection efficiency rates compared to HEK293 cells, which is why we chose this cell line. 

 Fig. 5C/D: label X-axis directly with FC compared to IgG control

We modified the x-axis on Figure 5 to clearly state it is compared to IgG control. 

 Fig. 7B: Coloring is confusing (looks the same as the AAV-syn-TDP-43)

We decided to remove this figure in agreement with the reviewer’s comment above. Please see our response above for further explanation.

Reviewer #2: The manuscript titled "Deciphering the Interactome of Ataxin-2 and TDP-43 in iPSC-Derived Neurons for Potential ALS Targets" by Tian et al. presents a valuable study on the interactome of TDP-43 and Ataxin-2. This research is of considerable interest to those studying ALS protein characterization, neurobiology, and related fields. I anticipate that this article will be widely cited. The manuscript is well-written, and I commend the authors for their efforts. I have only one minor suggestion for improvement: I recommend that the authors provide all raw data in a supplementary file along with a detailed description of their data analysis process, as this would be beneficial for readers.

We agree that providing raw data would add value to readers. We uploaded an Excel file of the raw data for the Mass Spectrometry experiment and have included a detailed methods section describing the analysis process. We also included a supplemental document showing uncropped Western blot data. Finally, all graphs in our manuscript have individual data points to represent the data. We hope that these efforts improve data transparency for our readers.

---

## [Editor Report · Decision Letter 1]

3 Dec 2024

Deciphering the interactome of Ataxin-2 and TDP-43 in iPSC-derived neurons for potential ALS targets

PONE-D-24-30547R1

Dear Dr. Tian,

We’re pleased to inform you that your manuscript has been judged scientifically suitable for publication and will be formally accepted for publication once it meets all outstanding technical requirements.

Kind regards,

Asif Ali

Academic Editor

PLOS ONE
---

## [Editor Report · Acceptance letter]

9 Dec 2024

PONE-D-24-30547R1 

PLOS ONE

Dear Dr. Tian, 

I'm pleased to inform you that your manuscript has been deemed suitable for publication in PLOS ONE. Congratulations! Your manuscript is now being handed over to our production team.

Kind regards, 

on behalf of

Dr. Asif Ali 

Academic Editor

PLOS ONE